# Deep Learning with Dynamic Computation Graphs

**Moshe Looks, Marcello Herreshoff, DeLesley Hutchins & Peter Norvig**
Google Inc.
`{madscience,marcelloh,delesley,pnorvig}@google.com`

## Abstract

Neural networks that compute over graph structures are a natural fit for problems in a variety of domains, including natural language (parse trees) and cheminformatics (molecular graphs). However, since the computation graph has a different shape and size for every input, such networks do not directly support batched training or inference. They are also difficult to implement in popular deep learning libraries, which are based on static data-flow graphs. We introduce a technique called dynamic batching, which not only batches together operations between different input graphs of dissimilar shape, but also between different nodes within a single input graph. The technique allows us to create static graphs, using popular libraries, that emulate dynamic computation graphs of arbitrary shape and size. We further present a high-level library[1] of compositional blocks that simplifies the creation of dynamic graph models. Using the library, we demonstrate concise and batch-wise parallel implementations for a variety of models from the literature.

## 1 Introduction

Training deep neural networks directly on minimally pre-processed corpora has led to many recent performance breakthroughs, mainly on problems in domains such as vision (Krizhevsky et al., 2012) and natural language (Bahdanau et al., 2015) where the inputs can be cast as dense $n$-dimensional arrays (henceforth *tensors*), or sequences of tensors. These successes exploit the effectiveness of training via gradient descent on mini-batches of tens to hundreds of inputs, implemented using the parallel SIMD capabilities of modern GPUs (Oh & Jung, 2004) and multi-core CPUs (Vanhoucke et al., 2011). This, in turn has led to a proliferation of libraries making it easier to train and deploy such models, by expressing them in terms of differentiable data-flow graphs over tensors (Abadi et al., 2016; Theano Development Team, 2016; Collobert et al., 2011).

However, there is also a long history of neural networks that compute over structures such as parse trees (Pollack, 1990), logical terms (Goller & Kuchler, 1996), and molecular graphs (Bianucci et al., 2000). In these models, each distinct input has a different computation graph structure; we say that they use *dynamic computation graphs* (DCGs). Such models continue to be developed and have recently yielded superior results on problems such as sentiment classification and semantic relatedness (Tai et al., 2015; Li et al., 2015), question-answering (Andreas et al., 2016), and screening of chemical compounds (Kearnes et al., 2016). Despite these successes, most practitioners avoid DCGs for implementation reasons. For example, Bowman et al. (2016) assert that "because TreeRNNs use a different model structure for each sentence ... efficient batching is impossible in standard implementations". Moreover, even if efficient batching were possible in principle, current libraries such as TensorFlow (Abadi et al., 2016) assume that the data-flow graph is static (i.e. is the same for each input) and impose a significant cost to graph construction, which makes it infeasible to build a new graph for each input.

Section 2 introduces dynamic batching, which enables efficient batching for training and inference with DCGs. Dynamic batching runs DCGs efficiently with existing libraries that only support static data-flow graphs; e.g. the same static graph can run a TreeRNN over any parse tree. We present empirical results for our implementation in TensorFlow. Section 3 presents a combinator library for concisely implementing models with DCGs using dynamic batching. Section 4 concludes.

---

[1] The library is called TensorFlow Fold and lives at `http://github.com/tensorflow/fold`.

## 2  DYNAMIC BATCHING

In deep learning libraries like TensorFlow, computations are manually batched. The computation is expressed as a static graph of mathematical operations, such as $y = \sigma(x \cdot w + c)$, which are polymorphic in batch size; an input $x$ of dimensions $(b, n)$ will yield an output of dimensions $(b, m)$, where $b$ is the batch size. With DCGs, the graph of operations is not static, but is assumed to be different for every input, so multiple inputs no longer naturally batch together in the same way. The *dynamic batching* algorithm overcomes this difficulty. Given a set of computation graphs as input, each of which has a different size and topology, it will rewrite the graphs by batching together all instances of the same operation that occur at the same depth in the graph. The rewriting process inserts additional `concat` and `gather` operations to move data between the batched operations; the indices to `gather` encode the topology of the original input graphs.

We distinguish between individual operations appearing as nodes in the underlying data-flow graph, such as addition or matrix-multiply, and small sub-graphs that conceptually act as functions over tensors, such as a feed-forward layer or LSTM cell. We refer to the former as "ops", and to the latter as "operations." Operations, (i.e. sub-graphs), form the building-blocks from which neural networks with DCGs are composed; dynamic batching schedules operations, not ops. Our algorithm requires that all operations which might be used be specified in advance, and it enumerates them for scheduling purposes. For example, a binary TreeRNN for NLP parse trees has two operations: embedding table lookups for words at the leaves of the tree, and RNN cells for the non-terminals.

The inputs and outputs of operations have *tensor types*. Each input or output may have a different type, but all types must be fixed and fully specified in advance. A tensor type consists of a *shape*, $x_1, \ldots x_n$, together with a scalar data type (e.g. `float32`). The inputs to an operation shall be tensors of dimension $(b, x_1, \ldots x_n)$, where $b$ is the batch size and $x_1 \ldots x_n$ is the shape of corresponding input tensor type. The outputs must all be tensors of dimension $(b, y_1, \ldots y_m)$, where $y_1, \ldots y_m$ is the shape of the corresponding output tensor type. Operations must be polymorphic with respect to the batch size, because the batch size will change each time the operation is invoked, depending on the topologies of the input graphs. However, their tensor types are fixed, so that it is possible to assign a known tensor type to each edge in the input computation graph.

The dynamic batching algorithm takes a directed acyclic computation graph as input. A batch of multiple input graphs can be treated as a single disconnected graph. Source nodes are constant tensors, and non-source nodes are operations. Edges connect one of the outputs of a node to one of the inputs of another node. Scheduling is performed using a greedy algorithm:

- Assign a depth to each node in the graph. Nodes with no dependencies (constants) are assigned depth zero. Nodes with only dependencies of depth zero are assigned depth one, nodes whose dependencies have a maximum depth of one get assigned depth two, etc.

- Insert pass-through (identity) operations so that an operation at depth $d + 1$ only refers to results at depth $d$.

- Batch together all nodes invoking the same operation at the same depth into a single node.

- Concatenate all outputs which have the same depth and tensor type. The order of concatenation corresponds to the order in which the dynamic batching operations were enumerated.

- Assign a label $(d, t, i)$ to each edge in the original graph, where $d$ is the depth, $t$ is the tensor type, and $i$ is the integer index for that edge into the (concatenated) outputs for $d$, $t$. The schedule for the graph consists of the indices $i$ for all edges, which are grouped together by depth and operation.

In our TensorFlow implementation, each dynamic operation is instantiated once in the static data-flow graph. The inputs to each operation are `tf.gather` ops, and the outputs are fed into `tf.concat` ops, as described above. These TensorFlow ops are then placed within a `tf.while_loop`. Each iteration of the loop will evaluate all of the operations at a particular depth. The loop maintains state variables for each tensor type $t$, and feeds the output of `concat` for tensor type $t$ and iteration $d$ into the input of the `gather`s at tensor type $t$ and iteration $d + 1$. The indices for `gather` at iteration $d$ are drawn from the edge labels $i$ for depth $d$ in the schedule. The initial values for the state variables at iteration/depth 0 are the constants in the input graph.

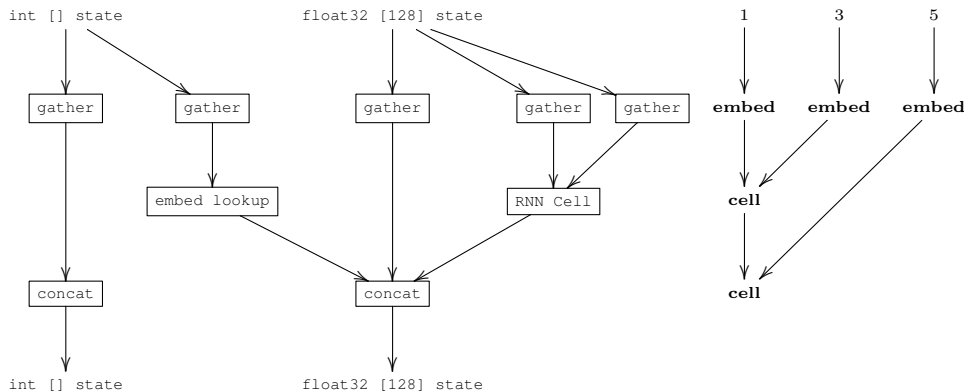

Figure 1: The static data-flow graph created by dynamic batching for a binary TreeRNN over parse trees (left), and input graph corresponding to the parse tree $((word_1, word_3), word_5)$ (right).

Dynamic batching allows us to construct a static TensorFlow graph that contains a single instance of each operation, yet can emulate input graphs of arbitrary size and topology where operations may appear an arbitrary number of times. The TensorFlow `concat`, `gather`, and `while_loop` ops are all differentiable, so gradients calculations and back-propagation do not require any additional code.

For example, a binary TreeRNN as described above yields a TensorFlow data-flow graph with a `tf.while_loop` whose body is shown on the left of Figure 1. Here each `gather` has an additional input (the indices for the given op at the given depth) which picks out which elements the operations are to be called with. The long downward arrows are the pass-throughs. The algorithm consumes a tree such as the one shown on the right of Figure 1 and turns it into inputs for the `gather` operations at each depth (here *depth* is the loop counter for the `tf.while_loop`.)

## 2.1 EXPERIMENTAL RESULTS

We have implemented dynamic batching as part of a new library, TensorFlow Fold, and designed a synthetic speed benchmark to compare it with manual batching in native TensorFlow. The benchmark uses the same underlying kernels and execution engine in both cases. Native TensorFlow cannot batch together trees of different shapes so, for testing purposes, we use a batch of random binary trees, all of which have the same shape. These test results thus represent a best-case scenario, in which all operations can be batched together perfectly. For the manual batching tests, we construct a static data-flow graph of operations corresponding to the shape of the tree. For the dynamic batching tests, we traverse each tree to construct a schedule, as described above.

The leaves of the tree are lookups into an embedding table, while the non-terminals implement a variant of the Tree-LSTM (Tai et al., 2015) equations. The tree size is 128, with a state size of 1024 for the LSTM. The CPU tests were run on a Dell z620 workstation with dual 8-core Intel Xeon processors (32 hardware threads), and the GPU tests were done using a consumer Nvidia GeForce GTX-1080 card. We compare manual batching, dynamic batching where all trees have the same shape, and dynamic batching where each tree has a different shape (the column marked "full dynamic"). There is no measurable penalty for dealing with trees of different shapes.

The test results shown in Table 1 emphasize the importance of batching, especially on GPUs. TensorFlow will launch a GPU kernel for every node in the tree, so there is a fixed overhead, proportional to the size of the tree, that dominates execution for small batch sizes. TensorFlow does not begin to saturate the GPU until relatively large batch sizes – 1024 or higher. The difference in speed between fully-batched and unbatched is over 160x.

Dynamic batching has less kernel invocation overhead because the data-flow graph is smaller. Dynamic batching instantiates each operation only once, and invokes it once for each depth, so the number of kernel invocations is $log(n)$, rather than $n$, where $n$ is tree size. Dynamic batching thus achieves substantial speedups even at batch size 1, because it batches operations at the same depth within a single tree.

Table 1: Inference timing benchmark; times are wall-clock averages in seconds

| batch-size | manual | | dynamic | | full dynamic | | cost | speedup |
|---|---|---|---|---|---|---|---|---|
| | batch | tree | batch | tree | batch | tree | ratio | ratio |
| (CPU) 1024 | 14.62 | 0.014 | 18.68 | 0.018 | 18.37 | 0.017 | 1.27 | 28.86 |
| 512 | 7.54 | 0.014 | 9.84 | 0.019 | 9.57 | 0.018 | 1.30 | 27.68 |
| 256 | 4.14 | 0.016 | 5.22 | 0.020 | 5.25 | 0.020 | 1.26 | 25.23 |
| 128 | 2.48 | 0.019 | 2.95 | 0.023 | 3.08 | 0.024 | 1.18 | 21.47 |
| 64 | 1.64 | 0.025 | 1.76 | 0.027 | 1.78 | 0.027 | 1.06 | 18.55 |
| 32 | 1.27 | 0.039 | 1.05 | 0.032 | 1.10 | 0.034 | 0.82 | 14.94 |
| 1 | 0.52 | 0.517 | 0.26 | 0.258 | 0.26 | 0.262 | 0.49 | 1.97 |
| (GPU) 1024 | 0.978 | 0.0009 | 1.590 | 0.0015 | 1.617 | 0.0015 | 1.62 | 101.79 |
| 512 | 0.530 | 0.0010 | 0.715 | 0.0013 | 0.721 | 0.0014 | 1.34 | 114.15 |
| 256 | 0.312 | 0.0012 | 0.323 | 0.0012 | 0.340 | 0.0013 | 1.03 | 120.86 |
| 128 | 0.236 | 0.0018 | 0.164 | 0.0012 | 0.178 | 0.0013 | 0.69 | 115.05 |
| 64 | 0.193 | 0.0030 | 0.093 | 0.0014 | 0.106 | 0.0016 | 0.48 | 96.40 |
| 32 | 0.153 | 0.0047 | 0.061 | 0.0019 | 0.074 | 0.0023 | 0.40 | 68.79 |
| 1 | 0.161 | 0.1608 | 0.038 | 0.0376 | 0.036 | 0.0359 | 0.23 | 4.47 |

However, the extra `concat` and `gather` ops that dynamic batching inserts do have a cost. The "cost ratio" column above shows the ratio between dynamic and manual batching, in the case where all trees in the batch have the same shape. The cost is only 20% for inference on GPUs with batch-size 1, but rises to 60% for training with backpropagation. The cost is mainly visible at large batch sizes, because it is balanced by the benefit of within-tree batching at smaller sizes.

Even with the cost, dynamic batching yields a 120x speedup over using a batch size of 1 on GPU, and 28x on CPU. The "speedup ratio" column above shows the ratio between the per-tree time for dynamic batching on random shapes ("full dynamic"), versus manual batching with a batch size of 1. Note that using a batch size of 1 is not actually feasible for TensorFlow, because TensorFlow has a large graph construction overhead, which is not included in these measurements, but it may apply to other libraries that lack such overhead.

## 3    A COMBINATOR LIBRARY FOR NEURAL NETWORKS

In addition to dynamic batching, the TensorFlow Fold library provides a set of combinators that simplify the task of constructing neural networks for DCGs. Our goal here is to show how dynamic batching enables implementing deep learning models (which are growing ever more complex) at a higher level of abstraction than manual batching. This in turn facilitates a more rapid feedback loop for trying out novel model variants, and thus obtaining superior results.

The design of the library was inspired by functional programming techniques such as *parser combinators* (Hutton & Meijer, 1996) and *arrows* (Hughes, 2000). In a combinator library computations are structured compositionally, by plugging together simpler computations in various ways. The basic unit of computation in TensorFlow Fold is a *block*, essentially a function from input to output. In a typical DCG model, the input is a graph or tree of some kind, and the output is a vector, which can be attached to a loss for training.

For example, consider a model where the inputs are sequences of words, of varying lengths, and the output is a sentence vector. Our library provide several different ways of handling sequences. Given a simpler block $f$ that operates on elements of the sequence, or $g$ on pairs of elements, we define the following combinators:

- `Map(f)`: yields $[f(x_1), f(x_2), \ldots f(x_n)]$. Applies $f$ to each element of the sequence, e.g. embedding each of the words of a sentence into $\mathbb{R}^N$.

- `Fold(g, z)`: yields $g(\ldots g(g(z, x_1), x_2), \ldots x_n)$. Applies $g$ sequentially in a leftward chain, e.g. running an RNN over a sequence. By default $z = 0$.

- `Reduce(g)`: yields $g(\texttt{Reduce}([x_1, \ldots x_{\lfloor n/2 \rfloor}]), \texttt{Reduce}([x_{\lfloor n/2 \rfloor + 1}, \ldots x_n]))$. Applies $g$ in a balanced tree,[2] e.g. max or sum-pooling over the elements.

Note that it is not necessary to pad or truncate sequences to the same length; dynamic batching handles sequences of differing lengths.

## 3.1 TYPE SYSTEM

Blocks are statically typed; each block has an input type and an output type. Types are inferred where possible, but must be explicitly specified in some cases. A type is one of the following:

- *Input* denotes objects in the host language (Python), such as trees and dictionaries.

- $Tensor_\texttt{dtype,shape}$ denotes tensors of a particular `dtype` and `shape`. [3]

- $Tuple(t_1, \ldots t_n)$, denotes a tuple of values of types $t_1, \ldots t_n$.

- $Sequence(t)$, denotes a sequence of elements of type $t$, of any length.

- *Void* is the unit type.

For example $Sequence(Sequence(Tuple(Tensor_\texttt{float32,[]}, Tensor_\texttt{int8,[3,4]})))$ denotes jagged arrays whose elements are pairs ($\texttt{float32}, \texttt{int8}^{3 \times 4}$).

## 3.2 BLOCKS AND COMBINATORS

Blocks are composed hierarchically; a block expression is always a tree. The non-terminals in the tree are combinators such as `Map` and `Fold`, which take simpler blocks as arguments. The leaves of the tree are atomic blocks, which include the following:

- `Scalar`: $Input \rightarrow Tensor$ Convert a Python scalar to a tensor.

- `Tensor`: $Input \rightarrow Tensor$ Convert a NumPy array to a tensor.

- `Function(h)`: $[Tensor \text{ or } Tuple(Tensor, \ldots)] \rightarrow [Tensor \text{ or } Tuple(Tensor, \ldots)]$
  Defines an operation $h$ (see Section 2) over tensors. Operations with multiple inputs and outputs use tuples of tensors.

- `InputTransform(h)`: $Input \rightarrow Input$
  Applies a user-defined Python function $h$ to pre-process the input.

In addition to the the sequence combinators described above, important combinators in the library include the following:

- $b_1 \texttt{ >> } b_2$: Function composition; the output of $b_1$ is fed to the input of $b_2$.

- `Record(`$\{l_1 : b_1 \,, \ldots \ l_n : b_n\}$`: $Input \rightarrow Tuple(t_1, \ldots t_n)$
  Takes a Python dictionary or tuple as input, and applies each block $b_i$ to the field labeled $l_i$, to yield an object of type $t_i$. Returns a tuple of the results for all fields.

- `OneOf(`$b_1, \ldots b_n$`)`: $Input \rightarrow t$
  Conditionally dispatches on its input to one of the blocks $b_1, \ldots b_n$.

- `Optional(`$b$`)`: $Input \rightarrow t$
  Applies $b$ if the input is not `None`, otherwise returns zeros. A special case of `OneOf`.

- `AllOf(`$b_1, \ldots b_n$`)`: $t_0 \rightarrow Tuple(t_1, \ldots t_n)$
  Passes its input of type $t_0$ to each of the blocks $b_1, \ldots b_n$, returning a tuple of results.

---

[2]`Reduce` uses a balanced tree rather than a chain in order to minimize computation depth and provide more opportunities for batching.

[3]Note that the leading batch size for tensors is not part of the `shape` of the corresponding *Tensor* type.

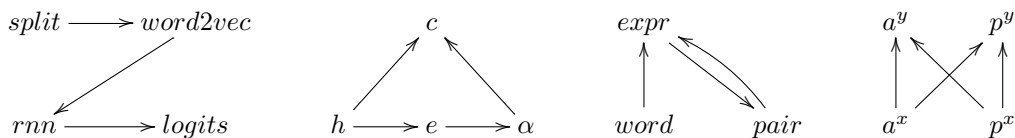

Figure 2: Block architectures for a pipeline (Section 3.3), feed-forward attention (Section 3.4), binary Tree-LSTMs (Section 3.5), and the weave module for molecule graphs (Section 3.6).

## 3.3 PIPELINES

Assume we have a set of (`text`, `label`) pairs as input and wish to predict the label from the text. The text consists of words, and we want to use an array of pretrained word embeddings (`word_matrix`) and corresponding dictionary mapping words to indices (`word_idx`). We call `word_idx.get(word)` to obtain the index of `word` in `word_matrix`, or `None` if `word` is unknown.

We start by creating a block which embeds each word into a continuous space:

```
word2vec = (InputTransform(word_idx.get) >>
            Optional(Scalar('int32')) >>
            Function(Embedding(initializer=word_matrix)))
```

This block uses an `InputTransform` to get the index of a word, which is passed to an `Optional` block that converts the scalar index to a tensor (or 0 if `None`). This in turn gets passed to an `Embedding` operation, which performs a lookup into an embedding table.

With `word2vec` in hand, we can define `text2vec`, which embeds sentences:

```
split = InputTransform(str.split)
rnn_cell = Concat() >> Function(FC(d, activation=tf.nn.relu))
text2vec = split >> Map(word2vec) >> Fold(rnn_cell, Zeros(d))
```

We use an `InputTransform` to split the string into words. Then we map the words to vectors with `word2vec`, and combine the word vectors with a simple RNN, which uses a single fully connected layer `FC` with $d$ hidden units. The `Zeros` block defines the initial state for the RNN.

Assume there are $n$ labels; we use a linear layer with $n$ outputs to get unscaled logits:

```
text2logits = text2vec >> Function(FC(n, activation=None))
```

For training, we create a `Record` block to convert the `label` to a tensor as well, and calculate loss:

```
record = Record([('text',  text2logits),
                 ('label', Scalar('int32'))])
loss = record >> Function(tf.nn.sparse_softmax_cross_entropy)
```

Finally, we create a `Compiler`, which validates a block, performs type-checking, and sets up dynamic batching in TensorFlow. Outputs of a compiled block are available as TensorFlow tensors, so training now proceeds as it would for any other TensorFlow model:

```
compiler = Compiler.create(loss)
cross_entropy = Compiler.output_tensors[0]
train_op = tf.train.AdamOptimizer().minimize(cross_entropy)
```

### 3.4 COMPLEX COMPOSITIONS

Recently, Raffel & Ellis (2016) have introduced an attention model for feed-forward neural networks. The model generalizes average-pooling and is defined as:

$$e_t = a(h_t), \alpha_t = \frac{\exp(e_t)}{\sum_{k=1}^{T} \exp(e_k)}, c = \sum_{t=1}^{T} \alpha_t h_t \tag{1}$$

where $a$ is a learnable function.

In this model, the block architecture is not a simple pipeline (i.e. a composition using `>>`) but instead forms a directed acyclic graph, as illustrated in Figure 2. A `Composition` block allows blocks to be composed into DAGs. The model code and details may be found in Appendix A.

### 3.5 RECURSIVE DEFINITIONS

$N$-ary Tree-LSTMs (Tai et al., 2015, sec. 3.2) generalize LSTMs from 1 to $N$ previous states. In Tai et al. (2015, sec. 5.1) they are applied to classify sentences from the Stanford Sentiment Treebank. This corpus consists of binarized constituency parse trees of one-sentence movie reviews, where every node has a sentiment label. At the leaves of the tree, words are mapped to word-embedding vectors which serve as the input to a binary tree-LSTM with 0 for the previous states. At the internal nodes, the LSTM takes 0 as input, and previous states from its two children. More formally,

$$h_{word} = TreeLSTM(Embedding(word), 0, 0) \tag{2}$$
$$h_{left,right} = TreeLSTM(0, h_{left}, h_{right}) \tag{3}$$

where $TreeLSTM(x, h_{left}, h_{right})$ is a learnable function corresponding to Tai et al. (2015) eqs. 9-14 with $N = 2$. Since a tree is a recursive data type, a model that processes trees must be recursively defined, as illustrated by the cycle in Figure 2. A `ForwardDeclaration` allows the creation of recursive models:

```
expr = ForwardDeclaration()
word = AllOf(Record([('word', word2vec)]),
             Zeros((state_size, state_size))
pair = AllOf(Zeros(embedding_size),
             Record([('left', expr()), ('right', expr())]))
expr_def = (OneOf(key_fn=len, case_blocks=[(1, word), (2, pair)]) >>
            TreeLSTM(state_size))
expr.resolve_to(expr_def)
```

A forward declaration like `expr` is not itself a block, but may be called (using the `expr()` syntax) to create references – i.e. blocks which refer to the declaration. The subsequent call to `resolve_to` then updates all the references to refer to `expr_def`.

The `word2vec` block is as defined in Section 3.3.

#### 3.5.1 EXPERIMENTAL RESULTS

Here we briefly report on some experiments with our implementation of $N$-ary Tree-LSTMs for sentiment analysis. While we set a new state-of-the-art, that is not really the point here. Our models are not particularly original, and could certainly be implemented without using TensorFlow Fold. What Fold does is to enable simpler and more concise definitions (see Table 3), along with faster execution, thus making it easier to rapidly explore novel model variants.

We used constituency Tree-LSTMs with tuned Glove vectors for word embedding, which achieved the best results of all sentiment models presented in Tai et al. (2015). In addition to this specific model, we have explored several novel variants.[4] In particular, Tai et al. (2015) employed non-

---

[4]Unsuccessful variants included standard LSTMs (i.e. having only a single forget gate) accepting pooled histories from their children, and models based on character rather than word-level embeddings.

Table 2: Test set accuracies on the Stanford Sentiment Treebank

| model | fine-grained | binary |
|---|---|---|
| Tai et al. (2015) | 51.0 (0.5) | 88.0 (0.3) |
| Munkhdalai & Yu (2016a) | 52.8 | 89.7 |
| Munkhdalai & Yu (2016b) | 53.1 | 89.3 |
| Ours (Single Model) | 52.3 (0.7) | 89.4 (0.4) |
| Ours (Ensemble) | **53.6** | **90.2** |

Table 3: Lines of code comparison

| model | ours | original | ratio |
|---|---|---|---|
| Feed-Forward Attention | 26 | 71 | 0.37 |
| Tree-LSTM | 119 | 219 | 0.54 |
| Graph Convolutions | 32 | 44 | 0.73 |

recurrent dropout and L2 weight regularization. We eliminated weight regularization in favor of the recurrent dropout scheme introduced by Semeniuta et al. (2016) and increased the LSTM state size from 150 to 300, leaving all other hyperparameters unchanged.

Results are shown in Table 2, including the best previously reported results. Fine-grained accuracy is measured for all trees and calculated based on the five possible labels. Binary accuracy is measured only for trees with non-neutral sentiment, and is based on negative vs. positive classification. The numbers in parentheses are standard deviations. Tai et al. (2015) report five independent runs, our results are based on thirty independent runs.[5] Noting the small size of this dataset (8544/1101/2210 trees for train/dev/test), we further evaluated an ensemble consisting of these thirty independently trained models; this variant sets a new state-of-the-art on both subtasks.

## 3.6 GRAPH CONVOLUTIONS

As a final example, we have used the Fold library to implement the graph convolution model introduced by Kearnes et al. (2016) for molecules, which are represented as undirected graphs of atoms. The code is more complex than our previous examples because it involves nested `Composition` blocks, and is given in Appendix B.

## 4 DISCUSSION

Neural architectures with dynamic computation graphs suffer from inefficient batching and poor tooling. Dynamic batching solves the former problem in full generality, we believe for the first time. The SPINN architecture (Bowman et al., 2016) is an alternative stack-based approach that also enables efficient batching with DCGs, but it is limited to binary trees, and requires padding/truncation to handle trees of different sizes. The Fold library addresses the tooling problem by providing a high-level combinator library which is intended to make it easy for practitioners to rapidly develop and iterate on architectures with DCGs.

The experimental results presented in section 2.1 quantify the impact of dynamic batching. The impact of the combinator library is harder to demonstrate quantitatively. One way to approach this (with a large grain of salt) is by comparing lines of code, which we do in Table 3, vs. the original author's sources. See Appendix C for details on the comparison protocol. Of course, a very short implementation is suboptimal if it comes at the cost of flexibility. The results in Section 3.5.1 show that models from the literature can be reimplemented in Fold, then extended to achieve superior performance. We suspect that other models with DCGs will have quite a bit of "head room" as well, due to simply having less work done tuning them compared with more mainstream architectures.

---

[5]Munkhdalai & Yu (2016a;b) do not report standard deviations or number of runs.

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

## A    FEED-FORWARD ATTENTION

The feed-forward attention model from Section 3.4 may be implemented in Fold as follows:

```
attention = Composition()
with attention.scope():
  h = attention.input
  exp_e = Map(a >> Function(tf.exp)).reads(h)
  z = (Sum() >> Broadcast()).reads(exp_e)
  alpha = ZipWith(Function(tf.div)).reads(exp_e, z)
  c = (ZipWith(Function(tf.mul)) >> Sum()).reads(alpha, h)
  attention.output.reads(c)
```

Within a composition `scope`, blocks may be wired together with `reads`, provided no directed cycles are formed. The `input` and `output` properties are used to define the overall inputs and outputs of the composition block. This example introduces several additional block types:

- `Sum` is a specialization of `Reduce` that performs elementwise addition.
- `ZipWith` is a variant of `Map` that accepts $n$ sequences as input and applies an $n$-ary function $f$ elementwise (stopping when the end of the shortest input sequence is reached).
- `Broadcast` creates a $Sequence(t)$ from a single $t$, repeating the same element endlessly.

## B    GRAPH CONVOLUTIONS

This section implements the graph convolution model introduced by Kearnes et al. (2016), for molecules represented as undirected graphs of atoms. There are real-valued feature vectors for each atom and for each distinct pair of atoms. For a molecule having $N$ atoms, we index its atom feature vectors as $a_i \in \mathbb{R}^n$ for $1 \le i \le N$. We index its pair feature vectors as $p_{i,j} \in \mathbb{R}^m$ for $1 \le i, j \le N$, where $p_{i,j} = p_{j,i}$ and $p_{i,i} = 0$.

The core of the graph convolution model is the weave module, which combines atom-level and pair-level features using six learnable functions (typically fully connected ReLU layers). The weave module can be stacked arbitrarily to create deep graph convolution models. Denoting inputs and outputs by $x$ and $y$ superscripts respectively, the weave module is:

$$a_i^y = f_A(f_{A \to A}(a_i^x), \sum_{j=1}^N f_{P \to A}(p_{i,j}^x)) \tag{4}$$

$$p_{i,j}^y = f_P(f_{A \to P}(a_i^x, a_j^x) + f_{A \to P}(a_j^x, a_i^x), f_{P \to P}(p_{i,j}^x)) \tag{5}$$

where $f_A, f_P, f_{A \to A}, f_{A \to P}, f_{P \to A}$ and $f_{P \to P}$ are learnable functions.

It is noteworthy that the $a^x \to p^y$ calculation involves a nested scan over the atoms; for each $a_i$ we must calculate $f_{A \to P}(a_i^x, a_j^x) + f_{A \to P}(a_j^x, a_i^x)$ for all $1 \le j \le N$:

```
a_i_to_p = Composition()
with a_i_to_p.scope():
  a_x_i = Broadcast().reads(a_i_to_p.input[0])
  a_x = a_i_to_p.input[1]
  f_i_j = ZipWith(Concat() >> f_a_p).reads(a_x_i, a_x)
  f_j_i = ZipWith(Concat() >> f_a_p).reads(a_x, a_x_i)
  p = ZipWith(Sum()).reads(f_i_j, f_j_i)
  a_i_to_p.output.reads(p)
```

The input to the `a_i_to_p` composition block is $(a_i^x, a_x)$. It has the type

$$Tuple(Tensor_{\texttt{float32,[n]}}, Sequence(Tensor_{\texttt{float32,[n]}})).$$

We broadcast $a_i^x$ over $a^x$ twice in succession to compute $f_{A \to P}(a_i^x, a_j^x)$ and $f_{A \to P}(a_j^x, a_i^x)$ for all $1 \le j \le N$, yielding `f_i_j` and `f_j_i`, which are length-$n$ sequences of vectors. We join and sum

each of these vectors elementwise to obtain the ultimate output of the block, which is also a length-$n$ sequence of vectors. The overall weave module may now be implemented as follows:

```
weave = Composition()
with weave.scope():
  a_x = weave.input[0]
  p_x = weave.input[1]
  a_to_a = Map(f_a_a).reads(a_x)
  p_to_a = Map(Map(f_p_a) >> Sum()).reads(p_x)
  a_y = ZipWith(Concat() >> f_a).reads(a_to_a, p_to_a)
  a_to_p = ZipWith(a_i_to_p).reads(a_x, Broadcast().reads(a_x))
  p_to_p = Map(Map(f_p_p)).reads(p_x)
  p_y = ZipWith(ZipWith(Concat() >> f_p)).reads(a_to_p, p_to_p)
  weave.output.reads(a_y, p_y)
```

The input to `weave` is $(a_x, p_x)$. It has the type

$$Tuple(Sequence(Tensor_{\texttt{float32,[n]}}), Sequence(Sequence(Tensor_{\texttt{float32,[m]}}))).$$

The calculation may be understood as follows:

- `a_to_a` maps over $a^x$ with $f_{A \rightarrow A}$, going from $Sequence(Tensor)$ to $Sequence(Tensor)$.

- `p_to_a` maps over $p^x$ with $f_{A \rightarrow P}$ and sums along the inner dimension, reducing from $Sequence(Sequence(Tensor))$ to $Sequence(Tensor)$.

- `a_y` zips `a_to_a` and `p_to_a` with $f_A$, going from
  $Tuple(Sequence(Tensor), Sequence(Tensor))$ to $Sequence(Tensor)$.

- `a_to_p` broadcasts $a^x$ over itself with `a_i_to_p`, expanding from $Sequence(Tensor)$ to $Sequence(Sequence(Tensor))$.

- `p_to_p` maps over $p^x$ with $f_{P \rightarrow P}$, going from $Sequence(Sequence(Tensor))$ to $Sequence(Sequence(Tensor))$.

- `p_y` zips `a_to_p` and `p_to_p` with $f_P$, going from
  $Tuple(Sequence(Sequence(Tensor)), Sequence(Sequence(Tensor)))$ to
  $Sequence(Sequence(Tensor))$.

## C CALCULATING LINES OF CODE

Our protocol for calculating lines[6] of code is as follows:

- Define the functional unit of comparison as an input-output mapping.
- Prepare a single file that implements this functionality and nothing else.
- Remove import statements, abstract base classes, logging, file i/o, and validation logic.
- Count lines of code, ignoring blank lines and comments.[7].

### FEED-FORWARD ATTENTION

The functional unit of comparison is creating the model for the variable-length experiment described in Raffel & Ellis (2016, sec. 2.3). This includes the loss and accuracy calculations, but does not include the training loop or the creation of training data. The original implementation[8] is in Python and uses Theano and Lasagne. The TensorFlow Fold implementation is more concise, partly due to differences between TensorFlow and Lasagne. Fold itself reduces implementation complexity by eliminating the need for manual batching, e.g. `x.sum(axis=1)` where batching is explicit over axis 0, vs. `x >> Sum()`, which is implicitly batched.

---

[6]All of the implementations we examine are formatted with 80-column lines excepting the Tree-LSTM implementation, which has a few lines that are slightly longer; we still count these as single lines.

[7]The calculations were performed with `cloc` (https://github.com/AlDanial/cloc).

[8]Commit `e8fce3e` from https://github.com/craffel/ff-attention.

Tree-LSTM

The functional unit of comparison is creating a (binary) constituency Tree-LSTM and running an epoch of training for the fine-grained sentiment classification task as described in Tai et al. (2015, sec. 5.1). This does not include loading the word embeddings or dataset, which are provided as inputs. The original implementation[9] is in Lua and uses Torch. Lua terminates blocks with the `end` keyword; we do not count these lines. Here, the use of Python and TensorFlow leads to substantially more concise code than with Lua and Torch. Unlike the previous example manual batching plays no role here, because the original implementation computes gradients and losses one tree at a time. Fold reduces complexity here by using a `OneOf` block to distinguish between leaves and internal nodes, rather than a recursive function that explicitly traverses the tree.

Graph convolution

The functional unit of comparison is creating a single weave module as described in Kearnes et al. (2016, sec. 3.3). The original implementation[10] is in Python and uses TensorFlow. Here, both implementations use the same language and deep learning library. Fold helps by eliminating the need for manual batching, as in the first example. This is particularly apparent in the atoms-to-pairs calculation, which requires making $n$ "copies" of an $n \times d$ matrix $x$ to get an $n \times n \times d$ tensor. In native TensorFlow the first dimension is batch, and the copying is explicit, as `reshape(tile(x, [1, n, 1]), [batch_size, n, n, d])`. In Fold, `x >> Broadcast()` suffices, because the number of copies needed is determined lazily by subsequent computations.

---

[9]Commit `b02ad49` from `https://github.com/stanfordnlp/treelstm`.
[10]Provided by Kearnes et al. (2016).

