# Peer review of "Deep Learning with Dynamic Computation Graphs"

_ICLR 2017 — accepted_

[Official Review · AnonReviewer1 · rating 8 · confidence 3 · 16 Dec 2016 (modified: 17 Jan 2017)]
**No Title**
soundness 5 · clarity 5 · impact 4

The paper presents a novel strategy to deal with dynamic computation graphs. They arise, when the computation is dynamically influenced by the input data, such as in LSTMs. The authors propose an `unrolling' strategy over the operations done at every step, which allows a new kind of batching of inputs.

The presented idea is novel and the results clearly indicate the potential of the approach. For the sake of clarity of the presentation I would drop parts of Section 3 ("A combinator library for neural networks") which presents technical details that are in general interesting, but do not help the understanding of the core idea of the paper. The presented experimental results on the "Stanford Sentiment Treebank" are in my opinion not supporting the claim of the paper, which is towards speed, than a little bit confusing. It is important to point out that even though the presented ensemble "[...] variant sets a new state-of-the-art on both subtasks" [p. 8], this is not due to the framework, not even due to the model (comp. lines 4 and 2 of Tab. 2), but probably, and this can only be speculated about, due to the ensemble averaging. I would appreciate a clearer argumentation in this respect.

Update on Jan. 17th:
after the authors update for their newest revision, I increase my rating to 8 due to the again improved, now very clear argumentation.

[Official Review · AnonReviewer2 · rating 8 · confidence 3 · 16 Dec 2016]
**A new method to optimize computation graphs**
soundness 5 · impact 5

The paper describes a novel technique to improve the efficiency of computation graphs in deep learning frameworks. An impressive speedup can be observed in their implementation within TensorFlow. The content is presented with sufficient clarity, although some more graphical illustrations could be useful. This work is relevant in order to achieve highest performance in neural network training.


Pros:

- significant speed improvements through dynamic batching
- source code provided


Cons:

- the effect on a large real-world (ASR, SMT) would allow the reader to put the improvements better into context
- presentation/vizualisation can be improved

[Official Review · AnonReviewer3 · rating 7 · confidence 5 · 19 Dec 2016]
**Description of a promising software package.**
originality 5 · clarity 5 · meaningful comparison 1

Authors describe implementation of TensorFlow Fold which allows one to run various computations without modifying computation graph. They achieve this by creating a generic scheduler as a TensorFlow computation graph, which can accept graph description as input and execute it.

They show clear benefits to this approach for tasks where computation changes for each datapoint, such as the case with TreeRNN.

In the experiments, they compare against having static batch (same graph structure repeated many times) and batch size 1.

The reason my score is 7 and not higher is because they do not provide comparison to the main alternative of their method -- someone could create a new TensorFlow graph for each dynamic batch. In other words, instead of using their graph as the scheduling algorithm, one could explicitly create each non-uniform batch as a TensorFlow graph, and run that using standard TensorFlow.

[Author Response · Moshe Looks · 13 Jan 2017]
**minor revisions in response to reviewer comments**

I've just uploaded a minor revision in response to reviewer comments, in particular clarifying the intent of sections 3 and 3.5.1 with new/revised initial paragraphs. This pushed us a bit over the page limit so I moved the code example from 3.4 into an appendix per AnonReviewer1's suggestion. Thanks again AnonReviewers for your time and attention, I think this version is certainly an improvement over the original draft.

[Author Response · Moshe Looks · 07 Feb 2017]
**uploaded minor revision**

Adds \iclrfinalcopy and a link to the github repo which is now live

[Final Decision · Program Chairs · 06 Feb 2017]
**ICLR committee final decision**

All reviewers viewed the paper favorably as a nice/helpful contribution to the implementation of this important class of methods.